# Music-to-Text Synaesthesia: Generating Descriptive Text from Music Recordings

## Abstract

In this paper, we consider a novel research problem, music-to-text synaesthesia. Different from the classical music tagging problem that classifies a music recording into pre-defined categories, the music-to-text synaesthesia aims to generate descriptive texts from music recordings for further understanding. Although this is a new and interesting application to the machine learning community, to our best knowledge, the existing music-related datasets do not contain the semantic descriptions on music recordings and cannot serve the music-to-text synaesthesia task. In light of this, we collect a new dataset that contains 1,955 aligned pairs of classical music recordings and text descriptions. Based on this, we build a computational model to generate sentences that can describe the content of the music recording. To tackle the highly non-discriminative classical music, we design a group topology-preservation loss in our computational model, which considers more samples as a group reference and preserves the relative topology among different samples. Extensive experimental results qualitatively and quantitatively demonstrate the effectiveness of our proposed model over five heuristics or pre-trained competitive methods and their variants on our collected dataset.[1]

## 1 Introduction

Multi-modal learning has drawn great attention in recent years and has been very developed in diverse applications, since our physical world is naturally composed of various modalities. Visual frames in videos are matched with text captions and these pairs have been widely used for video-language pre-training (Sun et al., 2019; Luo et al., 2021; Li et al., 2020); Kinects employ the RGB camera and the depth sensor for action recognition and human pose estimation (Shotton et al., 2011; Carreira & Zisserman, 2017); autonomous driving cars integrate the visible and invisible lights by the camera, radar and lidar for a series of driving-related tasks (Buehler et al., 2007; Torres et al., 2019); cross-modal retrieval aims to understand and match text with the existing textual repository and other modalities to meet users' queries (Nagrani et al., 2018; Suris et al., 2018; Zeng et al., 2021); language grounding learns the meaning of natural language meaning by leveraging the sensory data such as video or images (Bisk et al., 2020; Thomason et al., 2021).

Besides the above studies that employ multi-modal data to jointly achieve the learning task, translating information among different modalities, also known as synaesthesia, is another crucial task in the multi-modal community, where the text with its good compatibility and presentation ability, has become the intermediary of modality interaction. Various methods for synaesthesia between text and other modalities have been studied. Speech recognition can be directly regarded as a translation between the text and audio modality (Shen et al., 2018). Image captioning extracts the high-level visual cues and translates them into a descriptive sentence to describe the image content, while some studies consider the inverse process of image captioning by converting a semantic text into the visual image (Huang et al., 2021; Xu et al., 2015). Different from the existing modality translation studies, in this paper, we consider a novel problem, *the music-to-text synaesthesia*, i.e., generating descriptive texts from music recordings.

Recently, there are some pioneering attempts that build the connections between music recordings and tags at the initial stage. Cai et al. (2020) formulate music auto-tagging as a captioning task and

---

[1] Our code and data resources are available at `https://github.com/MusicTextSynaesthesia/MusicTextSynaesthesia`.

automatically outputs a sequence of tags given a clip of music. Zhang et al. (2020) use keywords of music key, meter and style to generate music descriptions, which can be used for caption generation. However, we argue that descriptive texts contain much richer information than tags, thus providing a better understanding of music recording. Moreover, we notice that tags might have a biased interpretation. Figure 1 presents two music recordings with the same music tags, but the opposite sentiment orientation of the text. The first one expresses a positive sentiment by describing the music as "*peaceful*" and "*beautiful*," while the second one uses tokens including "*sadness*" and "*loss*" to express a negative sentiment. It is clear that music tags are insufficient for describing the content of a music piece.

**Contributions**. In this paper, we propose a new task of generating descriptive text from music recordings. Specifically, given a music recording, we aim to build computational models that can generate sentences that can describe the content of the music recording, as well as the music's inherent sentiment. The major contributions are summarized as follows:

- From the research problem perspective, different from the music tagging problem, we propose the music-to-text synaesthesia, a cross-modality translation task that aims at converting a given music piece to a text description. To our best knowledge, the music-to-text synaesthesia is a novel research problem in the multi-modal learning community.

- From the dataset perspective, the existing music-related datasets do not contain the semantic description of music recordings. To build computational models for this task, we collect a new dataset that contains 1,955 aligned pairs of classical music recordings and text descriptions.

- From the technical perspective, we design a group topology-preservation loss in our computational model to tackle the non-discriminative music representation, which considers more data points as a group reference and preserves the relative topology among different nodes. Thus it can better align the music representations with the structure in text space.

- From the empirical evaluation, extensive experimental results demonstrate the effectiveness of our proposed model over five heuristics or pre-trained competitive methods and their variants on our collected dataset. We also provide several case studies for comparisons and elaborate the explorations on our group topology-preservation loss and some parameter analyses.

## 2 RELATED WORK

**Multi-modality Learning.** The goal of multi-modal machine learning is to build computational models that are able to process and relate information from different modalities, such as audio, text, and image. Baltrusaitis et al. (2019) describes five challenges for multi-modal learning, namely: *learning* representations, *translating*, *aligning*, *fusing*, and *co-learning* from/between different modalities. A large portion of prior works has focused on modality fusion, which aims at making predictions by joining information from two or more modalities. Applications include audio-visual speech recognition (Afouras et al., 2018), visual question answering (Goyal et al., 2017), emotion recognition, and media summarization. We now briefly review the literature and refer readers to Guo et al. (2019); Baltrusaitis et al. (2019) for more complete surveys.

**Translation between Modalities.** Beyond the applications of multi-modality translation we mention in the introduction section, here we introduce three common frameworks of multi-modality translation. (1) Encoder-decoder models directly learn intermediate representations used for projecting one modality into another. Zhang et al. (2017) adopt a sketch-refinement process to generate photo-realistic images for text-to-image synthesis tasks. Wang et al. (2021) designed a framework for end-to-end dense video captioning with parallel decoding. (2) Models with joint representations fuse multi-modal features by mapping representations of different modalities together into a shared semantic subspace. Sun et al. (2019) proposed ViLBERT, which extends BERT architecture to a multi-modal two-stream model, that learns task-agnostic joint representations of image content and natural language. Habibian et al. (2017) designed an embedding between video features and term vectors to learn the entire representation from freely available web videos and their descriptions. (3) Representations in coordinated representations based models exist in separated spaces, but are coordinated through a similarity function (e.g., Euclidean distance) or a structure constraint. These works include Wang et al. (2017), which presents a method to learn a common subspace based on adversarial learning for adversarial cross-modal retrieval. Peng et al. (2018) proposes modality-specific cross-modal similarity measurement approach for tasks including cross-modal retrieval. In

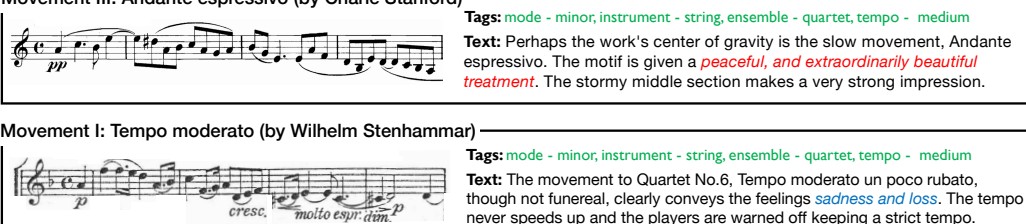

Figure 1: Samples of classical music and corresponding text descriptions in our collected dataset. The first piece is from String Quartet No.2 in A minor Op.45 composed by Charles Stanford, and the second is from String Quartet No.6 in D minor Op.35 by Wilhelm Stenhammar. While two samples have exactly the same music tags, it is clear that the two text descriptions have different sentiment.

this work, we experiment with different losses on the coordinate model, as it achieves the best performance among all three different types of models.

**Music Related Datasets and Tasks.** The existing music-related datasets mainly contain simple music tags. *AudioSet* dataset (Gemmeke et al., 2017) is a large-scale collection of human-labeled 10-second sound clips (not music recording) drawn from YouTube videos. This dataset only has descriptions for categories, not for individual sounds. The *MTG-Jamendo* dataset (Bogdanov et al., 2019) contains over 55,000 full audio tracks with 195 tags ranging from genre, instrument, and mood/theme categories. Oramas et al. (2016) describes a dataset containing reviewers from Amazon for albums. However, users' reviews may not necessarily describe the actual contents of music recordings. Cai et al. (2020) formulated the music tagging problem as a multi-class classification problem. A dataset called *MajorMiner* is used, with each music recording associated with tags collected from different users. Zhang et al. (2020) studied bidirectional music-sentence retrieval and generation tasks. The used dataset contains 16,257 folk songs paired with metadata information, including select key, meter and style as keywords. Text describing music recordings has limited style and mainly focuses on only some specific information.

We notice the studies on music captioning-related tasks address a similar research question (Choi et al., 2016; Gao et al., 2022; Manco et al., 2022). Specifically, Manco et al. (2021) used a private production music dataset, with their music clips of length between 30 and 360 seconds and captions contain between 3 and 22 tokens. Manco et al. (2021) also used an encoder-decoder network consisting of a multimodal CNN-LSTM encoder with temporal attention and an LSTM decoder.

## 3 DATA COLLECTION AND ANALYSIS

Since we address a new problem that generates descriptive texts from music recordings in this paper, a dataset containing aligned music-text pairs is required for model training. Although there are several public music/audio datasets with tags or user reviews (See Section 2), unfortunately, they do not meet our research problem for the following reasons: (1) From the text side, current datasets only have pre-defined tags for music pieces, rather than descriptive texts for music contents. (2) From the audio side, some clips are too short without a musical melody. In light of this, we collect a new dataset for the music-to-text synaesthesia.

**Data Collection and Post-Processing.** We collected the data from EARSENSE,[2] a website that hosts a database for chamber music, where Figure 1 shows an illustrative example of the music-text pairs. A typical music composition in chamber music contains several *movements*. Each movement has a title that normally contains tempo markings or terms such as minuet and trio; in some cases, it has a unique name speaking to the larger story of the entire work. As movements have their own form, key, and mood, and often contain a complete resolution or ending, we will treat each movement as the basic unit in this work. For example, Ludwig van Beethoven's Piano Sonata No. 8 in C minor, Op. 13 (also referred as *Sonata Pathétique*) contains three movements: (I) Grave (slowly, with solemnity), (II) Adagio cantabile (slowly, in a singing style), and (III) Rondo: Allegro (quickly). EARSENSE also provides comprehensive meta information for each music composition, including

---

[2]http://earsense.org/

composers, works and related multi-media resources. There is also an associated introductory article from professional experts, with detailed explanations, comments or analyses for movements.

By dividing the music compositions into movements according to their meta information, we collected 2,380 text descriptions in total, where 1,955 descriptions have corresponding music pieces. Each movement has its specific title, such as in-domain words indicating the types of music. This kind of information is useful when constructing a model as classical music follows some fixed patterns when composed. In general, tempo markings are specific but may differ from countries. In this work, we refer to Wikipedia[3] and convert Italic, French, German and other tempo markings into a universal four categories from slow to super fast. These categories are then added for movements as tags, by directly checking whether it contains tokens in our list.

**Preliminary Exploration.** The lengths of the 95% collected music pieces vary from 2.5min to 14min, which correspond to the descriptive texts with 14 to 192 tokens. The longest music length is up to 37 mins. We provide more details of these statistics in the Appendix.

Beyond the above basic statistics, we further explore the pairwise similarity and check whether similar music pieces share similar textual descriptions. Figure 2 shows the pairwise similarity matrix of music and text, respectively, where we use the cosine similarity on the music rep-

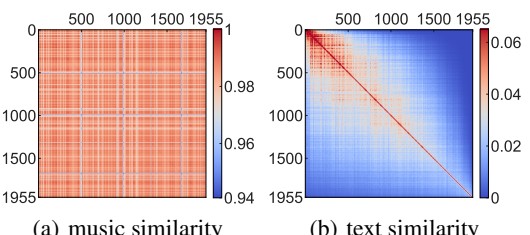

(a) music similarity      (b) text similarity

Figure 2: Pairwise similarity matrices of music representation by a self-reconstruction autoencoder and raw text by cosine and BLEU score.

resentations derived from a self-reconstruction auto-encoder and use BLEU score to calculate the raw text similarity. We notice that the texts are highly discriminative, where 96% of the pairwise similarities are below 0.06. However, all music representations have over 0.95 similarity due to the high similarities among classical music. It inevitably increases the difficulty of learning a mapping function for the music-to-text synaesthesia task that the similar music pieces match divergent texts. Thus, a natural idea is to employ the text to guide the learnt music representations to make them more discriminative, so the cross-modality transformation would be easier. The research question we explore in this paper is: *How can we employ the music-text pairs to learn the informative representation for music pieces and achieve the music-to-text synaesthesia?*

## 4  MODEL

The music-to-text synaesthesia is a cross-modality translation problem. Given $n$ paired tuple $\langle m_i, t_i \rangle$ with $1 \le i \le n$, where $m_i$ denotes the tuple $i$'s music and $t_i$ for its corresponding text, a music-to-text synaesthesia model $\mathcal{M} \to \mathcal{T}$ builds a mapping from the music to the text space, where $\mathcal{M}$ and $\mathcal{T}$ present the music and text space, respectively. In this section, we first provide a general overview of our model structure, followed by our proposed group topology-preservation loss to tackle the non-discriminative music representation.

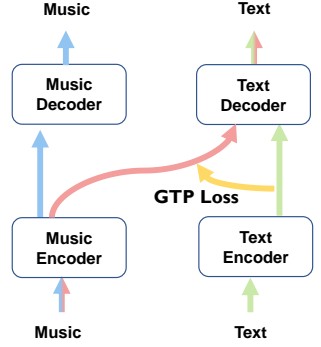

Figure 3: Framework for our music-to-text synaesthesia task.

**Cross-Modality Translation Model.**  Figure 3 shows our translation model, which is mainly based on the coordinate model (Peng et al., 2018) and our proposed group topology-preservation loss to achieve the translation between two modalities by learning the music/text latent space and their mapping by using three independent auto-encoders.

*Learning Music Representation.* We choose a convolutional neural network structure as our music feature extractor. Specifically, we use the spectrum of the raw music as inputs, where a CNN encoder and a transposed-CNN decoder are connected for the music reconstruction task. Let $f(\cdot)$ as the music encoder and $f'(\cdot)$ as the music decoder. We have the loss function for music representation

---

[3]https://en.wikipedia.org/wiki/Tempo

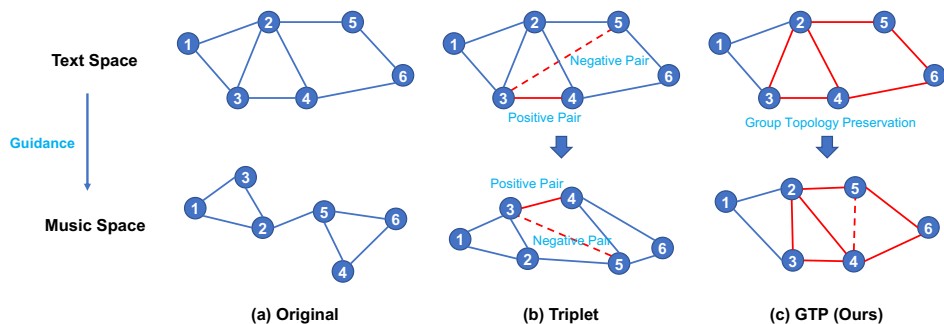

Figure 4: Our group topology-preservation (GTP) loss. (a) The structure of data points in the original text and music spaces. (b) The triplet loss is able to enforce the similarity between pairs of points, but fails to maintain the relative positions with other data points. (c) Our proposed GTP loss considers a group reference for each point and aligns the music representations with their structure in text space.

learning as follows:

$$\ell_{m2m} = ||m_i - f'(f(m_i))||^2. \tag{1}$$

*Learning Text Representation.* Similarly, in the text field, a transformer-based encoder and decoder are combined to train a text reconstruction model. Let $g(\cdot)$ be the text encoder and $g'(\cdot)$ be the text decoder. We have the loss function for text representation learning as follows:

$$\ell_{t2t} = \text{CrossEntropy}(t_i, g'(g(t_i))). \tag{2}$$

*Transforming From Music to Text.* With the above music and text representation, an extra auto-encoder is employed to achieve the mapping $\mathcal{M}$ from music to text.

$$\ell_{m2t} = ||f(m_i) - \mathcal{M}(g(t_i))||^2. \tag{3}$$

**Group Topology-Preservation (GTP) Loss.** Although the above coordinate model can achieve the music-to-text transformation, we notice that the music representations by the auto-encoder suffer from high non-discrimination, which further prevents the effective transformation. In light of this, we design a novel Group Topology-Preservation (GTP) loss to learn the discriminative music representation. Due to the relatively high discrimination in the text space, we aim to employ the text to guide the music representation generation.

To better illustrate our point, we provide a visualization of our designed group topology-preservation loss and the comparison with the triplet loss in Figure 4, where (a) shows the original representation space of music and text derived from the auto-encoders, (b) shows the effect of the triplet loss and (c) shows the effect of our GTP loss. When applying the triplet loss to node #3 in Figure 4(b), we observe that nodes #3 and #4 get close and nodes #3 and #5 are more separated. However, its relative positions as opposed to other data points are still different from the structure in the original text space. In Figure 4(c), our proposed GTP loss considers more data points as a group reference and preserves the relative topology among different nodes, thus it can better align the music representations with the structure in text space.

Formally, given a paired tuple $\langle m_i, t_i \rangle$ and a batch $\mathcal{B} = \{\langle m_j, t_j \rangle\}_{j=1}^{k-1}$ containing other $k-1$ paired tuples as the group reference, our GTP loss can be defined as follows:

$$\ell_{\text{GTP}} = ||\text{softmax}(\text{CON}(\cos(f(m_i), f(m_j)))_{j=1}^{k-1} - \text{softmax}(\text{CON}(\text{BLEU}(t_i, t_j))_{j=1}^{k-1})||^2, \tag{4}$$

where cos is the cosine similarity and BLEU score for the music and text's similarity calculation and CON is a concatenation function that flats the input elements and returns a row vector, and softmax is a normalizer across all elements. Our GTP enforces the given tuple and its group reference share the similar topology structure in both the hidden music space and the original text space, further separating the highly similar music representations. Moreover, the GTP loss not only captures the relationship between the given tuple and other samples in the group reference but also considers the relationship within the group reference.

**Overall Objective Function.** Our overall objective function can be written as follows:

$$\ell = \sum_{i=1}^{n} (\ell_{m2m} + \ell_{t2t} + \ell_{m2t} + \alpha\ell_{\text{GTP}}) + \beta\Omega(\Theta), \tag{5}$$

where $\Theta$ is the whole learning parameters including $f$, $f'$, $g$, $g'$ and $\mathcal{M}$, $\Omega$ is the parameter regularizer and $\alpha$, $\beta$ are trade-off hyper parameters among different losses in the objective function.

## 5 EXPERIMENTS

### 5.1 EXPERIMENTAL SETUP

**Comparison Models**. Here we propose five competitive methods and their variants for the music-to-text synaesthesia, which are in two main categories, heuristics and pre-trained methods.

- *Heuristics methods*. In our collected dataset, each music recording is associated with 4 types of tags, including mode, instrument, tempo and ensemble (see Table A2 in the Appendix). We note these tags normally contain information that categorizes the characteristics of the recordings. Motivated by this observation, we design two simple baselines that are based on heuristics. TAGS ($k$NN): Inspired by the $k$ nearest neighbor clustering, TAGS ($k$NN) assigns the text from the closest tags for each piece of music. Specifically, we directly use the text description from a music recording which has the largest number of same tags with our given music. TAGS (REPRESENTATIVE): Alternatively, we first divide all music pieces into separated groups based on four categories of tags. For each tag combination, we select a representative sentence, which has the highest BLEU score within its equivalent classes. These representative sentences can be formed as a text description for given music with the same tag combination.

- *Pre-trained methods*. ENCODER DECODER. The Encoder-Decoder model consists of a music CNN encoder and a text transformer decoder. The CNN encoder takes the spectrum of music as input and its output is then directly sent to the text transformer decoder to generate text descriptions. JOINT. The Joint Model is based on the assumption that information from different modalities is the same. A music encoder and text encoder are used to first encode the music and text inputs. Then the encoded outputs are sent to an adversarial discriminator to identify the encoding source, forcing the alignment of representation from different modalities. Finally, the encoding vector from both music or text can be decoded into text description without modality source information. COORDINATE. Coordinate model representing different modalities in different spaces. To do so, two autoencoders are used for music and text representation. And the text decoder is also required to learn a mapping of music to the description text. Moreover, based on COORDINATE, we propose its variants with additional losses for better music representation learning. Contrastive loss(Chen et al., 2020) builds positive samples by pitch shift, time stretch, randomly mask and noise addition(Won et al., 2020) to increase representation robustness. Pairwise loss pulls a pair of music representations together with similar texts. And triplet loss(Schroff et al., 2015) is an improved version that set a margin to pull similar samples and push other negative samples away. Figure A3 in the Appendix shows the differences among the Encoder Decoder, Joint Model and Coordinate Model.

**Implementation**. From the music side, the original classical music is first converted into a spectrogram which has a frequency axis of 256 dimensions and a flexible temporal axis. It is then encoded into $T \times 768$ by CNN encoder, where $T$ is the music length in seconds. Both the music encoder and decoder contain 8 layers. From the text side, the text descriptions are tokenized by BERT tokenizer and encoded into $L \times 768$, where $L$ is the number of tokens. The text encoder and decoder consist of 5 layers. As the input music and text are of varying length, both music and text encodings are divided into 20 segments evenly, and mean pooled into $20 \times 768$ vectors as a fixed-length representation. The same number of layers for music and text encoders are used for Encoder-Decoder and Joint Model baselines.

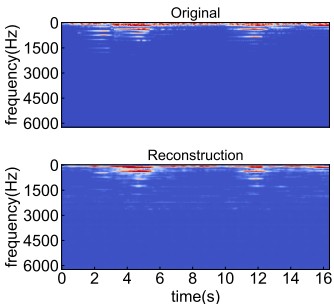

Figure 5: Spectrum diagrams from the original music (top) and reconstruction (bottom) of our pre-trained music autoencoder.

For our model, we first pre-train music and text auto-encoders on our constructed dataset. To prevent data leakage, the text auto-encoder is pre-trained without the test data. Moreover, these music and text auto-encoders capture a good representation in the reconstruction task. Figure 5 shows the spectrum diagrams of the original and reconstruction music from music representation, which contains the main information of original music. Meanwhile, the text-autoencoder reconstructs from

representation with an average BLEU score of 79.66. During training, we set the learning rate to be $5e^{-5}$ and the batch size is 8, implemented by gradient accumulation. The hyperparameters in our model are set to be $\alpha = 500$, $\beta = 5$ and $k = 32$. Considering the computational budget, our $k$ grouped sample representations are implemented by a queue according to MOCO (He et al., 2020), where we employ a momentum encoder with $momentum = 0.999$.

**Evaluation**. As the size of our dataset is rather small, we evaluate the model performance using the 5-fold cross-validation for all our experiments. We use the BLEU score (Papineni et al., 2002) as our main evaluation metric to compare our generated captions with the reference text descriptions. BLEU score has been confirmed to have a high correlation with human judgements, and it is widely used in natural language generation tasks, such as machine translation and text summarization.[4]

## 5.2 MAIN RESULTS

For each data split, we run 5 experimental trails and then report the average and standard deviation of the BLEU scores in Table 1. Note that the ground text descriptions for classical music are sometimes hard to digest even for humans (see the cases in Table 2), it is thus not totally surprising that the performance of our model is generally low.[5]

We observe that heuristics baseline methods have lower performance compared to learning-based models. It is intuitive as heuristics-based methods are not able to generate text descriptions that are adjusted to given music contents. Among all three representative models for multi-modal tasks, COORDINATE model achieves the best performance with an average BLEU score of 6.42. When further adding different losses to the training objective of the coordinate model, we observe similar performance for using triplet loss and even drops for contrastive loss and pairwise loss; while there is a significant performance improvement when using our proposed GTP loss, achieving the highest BLEU score of 6.76. Our experimental results compared to baseline methods are statistical significant with $p$-value $< 0.001$. Some music recordings on earsense have a second comment by another commentator. We collect these data to be another dataset with 50 records and validate our models on it. The BLEU score of the baseline Coordinate model is 8.54 and our GTP loss model is 8.81.

To better interpret this experimental result, in Figure 6 we plot the music and text similarities of a pair of different music-text tuples along with the histograms on the music similarity when applying different losses. We observe that the output representations from pre-trained music autoencoder have high similarity, with most above 0.90 in Figure 6(a). It is because the high frequency of all music is almost blank while classical music also has a similar low frequency with several limited instruments. When using pair loss, we observe the music representations simply gather together and become indistinguishable (shown in Figure 6(b)). Meanwhile, using our proposed GTP loss helps maintain a structure among music representations. As shown in Figure 6(c), the music similarities vary from -0.05 to 0.4 and the range is much wider than that of other models. Also, the representations spread out and become more distinguishable, which naturally helps the performance of text generation. It validates the motivation of our designed GTP loss that when considering more data points as a group reference, the model is able to capture the relative topology among different nodes, thus can better accomplish the transformation from different modalities.

We further present several examples in Table 2, showing the comparisons between ground truth music descriptions and the generated ones from COORDINATE with or without our proposed GTP loss. We observe that COORDINATE and OURS are able to identify basic characteristics of the input musics, while our model in general could capture more subtle differences. For example, in Case #1, both models correctly identify the tempo category "*allegro*," while our model further captures the music instrument of "*cello*" and genre of "*sonate.*" In Case #2, our model correctly captures two themes in one movement, where the "*bright and lively*" first theme is recognized as a "*march like a scherzo,*" and the "*dreamy*" second theme is described to be "*romantic*".

It can be observed that, for objective facts, models can simply and correctly tell the correct term, while our model can further capture more details. When it comes to subjective description, our model could capture the sentiment orientation of music themes, although it is still a challenging task

---

[4]The typical BLEU score for machine translation task is from 10 to 40 (Vaswani et al., 2017).

[5]We do not report the performance of MusCaps (Manco et al., 2021) due to their low performance. We conjecture it comes from the different music feature backbone.

Table 1: Comparisons of different models for our music-to-text generation task using BLEU metric. For all neural-based methods, we run experiments 5 times for each split and then report average scores and standard deviations. $k$ in triplet loss refers to different pair numbers used.

| | Split #1 | Split #2 | Split #3 | Split #4 | Split #5 | **Average** |
|---|---|---|---|---|---|---|
| TAGS ($k$NN) | 3.95 | 3.73 | 3.6 | 3.84 | 3.57 | 3.74 |
| TAGS (REPRESENT.) | 3.75 | 3.65 | 4.06 | 3.51 | 3.13 | 3.62 |
| ENCODER DECODER | $5.86_{\pm 0.21}$ | $6.31_{\pm 0.27}$ | $6.36_{\pm 0.15}$ | $5.77_{\pm 0.20}$ | $6.59_{\pm 0.30}$ | 6.18 |
| JOINT | $5.84_{\pm 0.66}$ | $6.31_{\pm 0.24}$ | $6.48_{\pm 0.53}$ | $5.83_{\pm 0.27}$ | $6.27_{\pm 0.27}$ | 6.15 |
| COORDINATE | $6.41_{\pm 0.34}$ | $6.70_{\pm 0.17}$ | $6.20_{\pm 0.21}$ | $6.10_{\pm 0.11}$ | $6.66_{\pm 0.24}$ | 6.42 |
| COOR. + Contrastive loss | $5.71_{\pm 0.16}$ | $4.80_{\pm 0.18}$ | $6.74_{\pm 0.24}$ | $6.11_{\pm 0.11}$ | $6.98_{\pm 0.47}$ | 6.07 |
| COOR. + Pairwise loss | $6.20_{\pm 0.32}$ | $6.76_{\pm 0.10}$ | $6.53_{\pm 0.29}$ | $5.97_{\pm 0.20}$ | $6.53_{\pm 0.21}$ | 6.40 |
| COOR. + Triplet loss ($k$=1) | $6.60_{\pm 0.15}$ | $6.47_{\pm 0.13}$ | $6.74_{\pm 0.07}$ | $6.07_{\pm 0.13}$ | $6.95_{\pm 0.14}$ | 6.56 |
| COOR. + Triplet loss ($k$=32) | $6.36_{\pm 0.28}$ | $6.59_{\pm 0.07}$ | $6.50_{\pm 0.20}$ | $5.94_{\pm 0.24}$ | $6.71_{\pm 0.21}$ | 6.42 |
| OURS | $6.94_{\pm 0.19}$ | $6.71_{\pm 0.15}$ | $6.91_{\pm 0.25}$ | $6.37_{\pm 0.24}$ | $6.87_{\pm 0.16}$ | **6.76** |

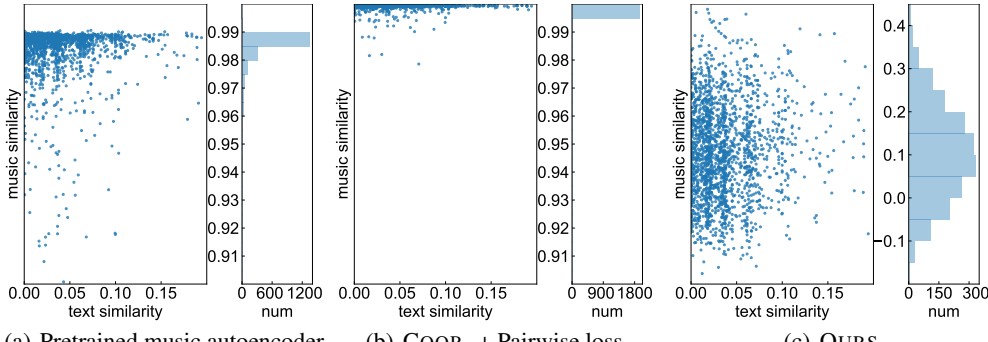

(a) Pretrained music autoencoder    (b) COOR. + Pairwise loss    (c) OURS

Figure 6: Music and text similarities of a pair of different music-text tuples and histograms on the music similarity, where the music similarity is calculated by the cosine similarity on the learned representation and the text similarity is calculated by the BLEU score on the original texts. (a) Pretrained music autoencoder; (b) COORDINATE with pairwise loss; and triplet loss has a similar histograms; (c) OURS model (with GTP loss).

due to no sentiment supervision and expression diversity (see Section 5.3 for further analyses of the sentiment transfer ability between different modalities). We also notice some grammar mistakes and fixed patterns appear in generated sentences. It seems that even with sufficient pre-training, it is still hard to learn a professional expression of both correctness and fluency from limited dataset.

## 5.3 IN-DEPTH ANALYSES

**Effects of Hyperparameters.** We are interested in the effect of the number $k$ of samples considered in our GTP loss. To do this, we vary the number of $k$ from 4 to 2,048 and plot the BLEU scores in Figure 7(a). We observe that the performance of our model first increases and achieves its peak when $k$ is chosen to be 32; the performance then drops when $k$ is larger. It provides empirical evidence that considering more information from other data points do help improve the model performance. However, a larger $k$ is not always better for which will make older samples in the momentum encoder queue more outdated. We observe a similar pattern when varying the scale of the hyperparameter $\alpha$ from 1e-2 to 1e5 in Figure 7(b). The model yields the best performance when $\alpha = 500$.

**Transfer Ability of Sentiment.** Since our task is music-to-text synaesthesia, we care about whether our model could learn a transfer capability from the music domain to the text domain. To explore this, we analyze whether the generated text descriptions are able to reflect the sentiment from the original ground truth text. We select a popular sentiment classifier[6] that could detect a variety of 7 sentiment categories and then apply this classifier to identify the sentiment in reference text and text descriptions generated by our model. Figure 7(c) shows the sentiment distribution of our generated texts and the original reference texts. We observe a skewed distribution that a majority of

---

[6] https://huggingface.co/j-hartmann/emotion-english-distilroberta-base

Table 2: Generated text descriptions between different models given a classical music piece, followed by their sentiments [joy/neutral/sadness] in the end. We use different colors to highlight phrases: blue for noun phrases that is objective facts and yellow for subjective adjective phrases.

| | |
|---|---|
| Case #1 | **Ref.**: the *rousing allegro* vivace finale begins with a *sweetly* singing adagio introduction that very briefly seems to recall the extended introductions of the earlier op. 5 sonatas and provides the only real hint of a slow movement that will otherwise have to wait for the last *cello sonata* for its fully independent flowering. [joy]
**COORDINATE**: the movement, *allegro* moderato, begins with a fugue with a fugue. [neutral]
**OURS**: the movement, *allegro* moderato, begins with a theme of the main theme of the main theme in the *cello*. the main theme is a kind of the middle section, the work of the movement. the music is a fugue with a bit more *lyrical* second theme to the *sonata* form of pizzicato. [joy] |
| Case #2 | **Ref.**: the movement, *allegro molto* e con brio, is based on two subjects, *the first is bright and lively* while *the second is dreamy* with an improvisational aura. the development is ingenious and an exciting coda caps off this first rate work. [joy]
**COORDINATE**: the movement, *allegro moderato*, begins with a theme in the main theme. [joy]
**OURS**: the movement, *allegro molto*, is a series of variations that it *begins with a march - like it is a scherzo* and a sense of the main theme, in the movement. it is a very *romantic second theme*, a sense of the movement. [joy] |
| Case #3 | **Ref.**: the movement, *allegro moderato*, begins with a *somewhat passionate, very characteristic* quartet - like melody which leads to an ingratiating, *lyrical* second theme. [joy]
**COORDINATE**: the movement, *allegro*, begins with a lengthy introduction. [neutral]
**OURS**: the movement, *allegro*, begins with a *lively allegro con brimming* with a long series of the main theme. it begins with a very *romantic* melody, the music of the movement. [joy] |

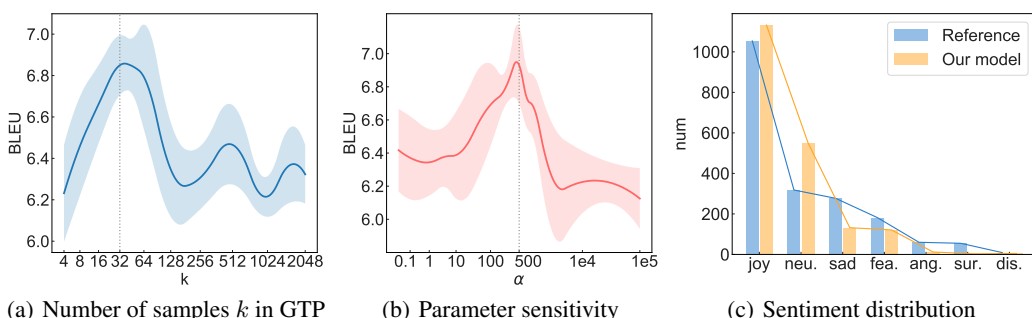

(a) Number of samples $k$ in GTP    (b) Parameter sensitivity    (c) Sentiment distribution

Figure 7: (a) BLEU score when varying the number of neighbors $k$. (b) Parameter sensitivity on $\alpha$. (c) Sentiment distribution of our generated texts and the original reference texts.

descriptions from both references and generated models are classified as "joy," while the left others are tagged as having sentiments like "neutral," "sadness," and "fear." These experimental results are consistent with our intuitions that classical music normally brings people with happiness and joy. We observe that the distributions for the sentiment labels from reference and our model are similar, with a Pearson correlation coefficient up to 0.96.

## 6    CONCLUSION

In this paper, we proposed a novel task of music-to-text synaesthesia, the goal of which is to generate text descriptions for a given music piece. As current existing datasets normally do not contain semantic descriptions for music pieces, we collected a new dataset that contains 1,955 classical music recordings with professional text descriptions. We also proposed a group topology-preservation loss that preserves relative group topology among music and text, given the fact that classical music is non-discriminative. We conducted experiments on five heuristics or pre-trained competitive baseline methods, based on existing multi-modal literature. Compared with other sample level constraint losses, our model with GTP loss extracted better music representation and generated better text descriptions. We further performed extensive experiments including case study and sentiment analysis to demonstrate the effectiveness of our proposed music-to-text synaesthesia model.

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

# A  DATA ANALYSIS

We summarize the statistics of our newly collected dataset in Tables A1 and A2,Figures A1 and A2. There are several advantages and unique features for this dataset: (1) Music pieces are semantically hard enough, which are provide space for generating text with rich information. (2) The descriptive texts are generated by music professionals, thus are freestyle and diverse.

Table A1: Statistics of our collected dataset

| | |
|---|---|
| # of unique music compositions | 786 |
| # of unique composers | 304 |
| # of music-text pairs | 2,380 |
| # avg. of tokens per text description | $45.7 \pm 41.2$ |
| # avg. of each music piece (seconds) | 305 |

Table A2: Four categories of tags in our dataset

| Tag | Type |
|---|---|
| mode | major, minor |
| instrument | string, wind, piano |
| tempo | slow, medium, fast, super faster |
| ensemble | sonate, trio, quartet, quintet or more |

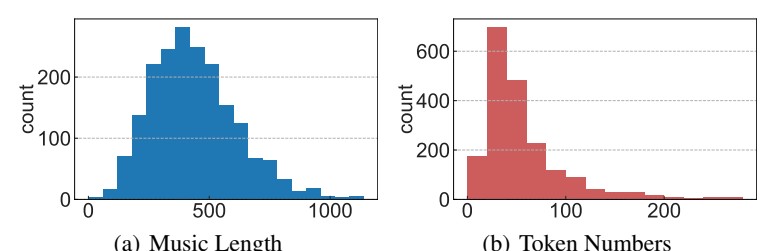

(a) Music Length      (b) Token Numbers

Figure A1: Distribution of music length and token nums

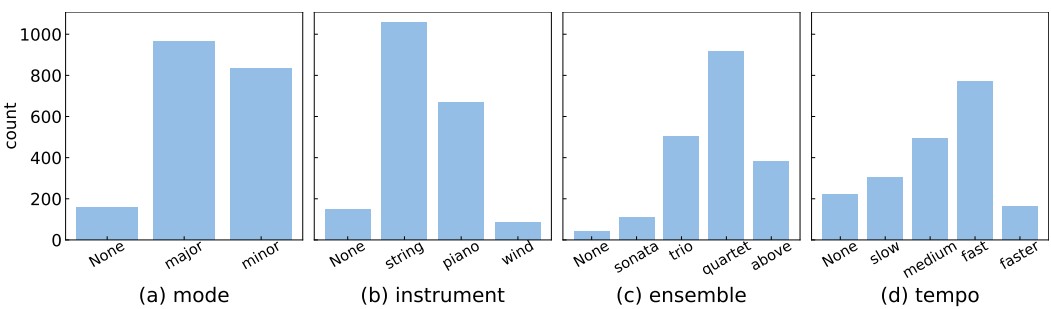

(a) mode    (b) instrument    (c) ensemble    (d) tempo

Figure A2: Distribution of four tag categories

# B  STRUCTURES OF BASELINE MODELS

We visualize the structure of all baseline models in Figure A3.

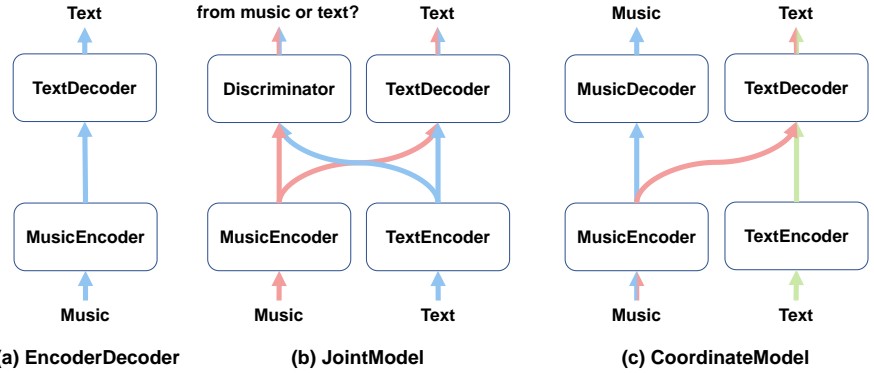

(a) EncoderDecoder    (b) JointModel    (c) CoordinateModel

Figure A3: Structures of baseline models.

