# OpenReview forum: "Music-to-Text Synaesthesia: Generating Descriptive Text from Music Recordings"
_ICLR.cc/2023/Conference — Submitted to ICLR 2023_

### Official Review · Reviewer_fsng · 2022-10-27

**Confidence:** 3
**Correctness:** 3
**Technical Novelty And Significance:** 3
**Empirical Novelty And Significance:** 3
**Recommendation:** 5

**Clarity, Quality, Novelty And Reproducibility:**

Clarify and quality: Paper is well written, experiment and evaluation are clear. The justification of GTP can be clarified better.
Novelty: Medium. Treating music captioning (or any captioning) as sequence-to-sequence (i.e., translation) problem exists. The proposed loss is interesting but of limited novelty.
Reproducibility: The proposed model is explained well but not completely for those who want to re-implement it.

**Strength And Weaknesses:**

Strength: The system and the experiment are well designed and executed. The evaluation seems to be done correctly, too.
Weakness: The system requires a paired dataset, which is hard to find outside of classical music. The literature review lacks some relevant works. Not calling the problem music captioning may be misleading.

**Summary Of The Paper:**

This paper introduces a music captioning system that was trained on a classical music dataset. A modification on loss is proposed to improve the performance.

**Summary Of The Review:**

Title - Is there any reason to avoid music captioning?
Introduction - Similarly, the recent progress in music captioning (and even audio captioning maybe) is missing.
Section 2 - I appreciate the preliminary exploration.
Section 3 - Good approach to see the problem a cross-modality translation problem. The proposed system and GTP loss seems sensible to me.
On GTP section
  - "by the autoencoder suffer from high non-discrimination.": Can you elaborate this?
  - "Due to the relatively high discrimination in.. ..generation.": I get it, but it'd be nicer if this is elaborated, too.
- Is "node" a music item?
Section 4
 - Choice of BLEU - It makes sense. But it would be even better if there's some example about how BLEU works for this problem / dataset.
Section 4.2
  - "We observed that the output representations ... ... (shown in figure 6(b))": But this is also because the design of the feature extractor or the similarity measure is not suitable. Not saying GTP loss is bad, but this does not mean it is absolutely necessary. Perhaps worth mentioning it somewhere to clarify.
- Under "Transfer Ability Sentiment": "These explanation results are consistent with our.. ..happiness and joy": I find it difficult to simply agree with. And how is it related to the proposed work? The output label distribution would follow the training data as long as the training set and evaluation set are stratified.

---

> ### Author Response · Authors · 2022-11-17
> **Response to Review fsng**
>
> Thanks for your detailed review and generous comments. Below we address our main points regarding this review.
>
> **The system requires a paired dataset, which is hard to find outside of classical music.**
> We agree that a paired dataset is hard to find, and it is indeed our challenge to train such systems.
> In this work, we make our best attempt to build music-text pairs using https://www.earsense.org/.
> We hope our work would draw people's attention on this cross-modality task and get collective efforts from the community to build these music-text pairs.
>
> **Not calling the problem captioning may be misleading.**
> At the very beginning, we choose "music captioning" as the candidate, which makes the reader easier to get the point due to the well-known image captioning. However, we notice that the generative texts in image captioning are usually short. Inspired by a famous prose *Moonlight over the Lotus Pond*, where the author used synaesthesia to build the connection among vision, music, and text, we use the term "synaesthesia" to define our task. But we are also open-minded to using "music captioning" instead, if this is more exact and understandable. Please advise.
>
> **Can you "elaborate by the autoencoder suffer from high non-discrimination"** The music representation obtained by the autoencoder has high cosine similarity to each other, which means the pre-trained music encoder pays little attention to the discriminated information in the music and affects the music-text transformation.
>
> **Is "node" a music item?** Nodes above are text items and nodes below are music items. The GTP loss uses text topology to guide the music representation.
>
> **Feature extractor**. We agree with Reviewer fsng that the music representation is not good, which has been demonstrated in Section 2.
>
> Using a pre-trained music model on other large music corpus makes our task easier, which actually was one of our attempts at the beginning of this project. We have considered using pre-trained music tagging models [1] to extract music features. Unfortunately, the generated descriptions were much worse with all the baseline models. We conjecture that these CNN-based models are not suitable for our collected classical music.  Thus, we trained an auto-encoder to check the representation quality through reconstruction. The average music length in our dataset is 7.4 mins, which means the auto-encoder was trained in 242 hours of music pieces. Even though the pre-trained music model is not good, it is the best we can achieve so far. If Reviewer fsng has some ideas on a better feature extractor, we are eager and happy to have a try.
>
> **How is sentiment analysis reasonable and related to the proposed work?** We believe the paired music and text should have the same sentiment. Although our proposed models do not consider sentiment explicitly during the training, we are interested to see whether our model has the capacity for sentiment transformation.
>
>
> **Choice of BLEU**.
> BLEU score measures the similarity of a given text to the reference text, and it has been confirmed to have a high correlation with human judgments.
> Due to its effectiveness and cheapness to compute, it is widely used in natural language generation tasks, such as machine translation and text summarization.
> As our goal is to evaluate the quality of the generated music descriptions with the ground truth, we select the automatic BLEU score as our metric. The typical range of BLEU scores for machine translation tasks is from 10 to 40 [1].
>
>
>
> [1] Ashish Vaswani, Noam Shazeer, Niki Parmar, Jakob Uszkoreit, Llion Jones, Aidan N. Gomez, Łukasz Kaiser, and Illia Polosukhin. 2017. "Attention is all you need", Proceedings of the 31st International Conference on Neural Information Processing Systems. 2017.

---

### Official Review · Reviewer_DjXC · 2022-10-29

**Confidence:** 3
**Correctness:** 2
**Technical Novelty And Significance:** 2
**Empirical Novelty And Significance:** 1
**Recommendation:** 3

**Clarity, Quality, Novelty And Reproducibility:**

The presentation was reasonably clear and there would not be too much difficulty in reproducing most of the results.  However it is a bit lacking in terms of novelty and substance.

**Strength And Weaknesses:**

The authors begin by laying out an awkward case, defining translating information between modalities as synaesthesia, and therefore claiming that speech recognition and image captioning as forms of synaesthesia.  I don't want to argue the particular semantics of this, only to say that if synaesthesia here is going to be applied so broadly, it loses any important meaning and misleads the reader.  Being familiar with the definitions of synaesthesia as a -perceptual- phenomenon, I certainly thought there would be deeper cognitive connections, so felt a bit mislead by the title. Alternatively, in terms of modeling, I would have expected something more along the lines of [2], where more emphasis is put on a shared latent space (not unlike earlier sorts of image captioning - Socher, etc.).

But if image captioning is a form of synaesthesia, then there is another name for such research that springs to mind: music captioning.  Looking down this line of research, a number of related work pops up [1-4].  And in terms of music labeling vs. music "synaesthesia", if the only difference between these is whether an output vector is mapped to one label from a set of many, or decoded into many words, it seems a weak point from which to pitch this research as having an important distinction from other related audio->text tasks.  So while I think the general idea of learning to predict textual descriptions from raw music audio is interesting and should be pursued, the presentation of this paper falls short in terms of recognizing existing work, or establishing a novel task.

It terms of the dataset, it seems useful but is limited in both size and the domain of being a specific set of classical music pieces.  On the text side, the descriptions (as viewed through the generating text descriptions) do not seem that diverse nor bring something by virtue of being full sentences that could not have been conveyed by bag of words.

The experimental setup is reasonable given the lack of established baselines for this type of task, yet, they also seem weak enough or inherently disadvantaged that they serve little competitive function.  The two experiments that are more interesting are with/without the topographic loss modifications.  The GTP loss itself seems well-motivated and may be only applicable to datasets of this sort, but stands out as one of the novel contributions.  However, when we see the generated text descriptions in Table 2, it led me to believe the learning problem on this data is quite simple, and might be the classification of just a few categories, strung together with text, and much less of a newly established task where the text descriptions themselves offer something qualitatively more informative than a multi-label classification task.


[1]
@article{Choi2016TowardsMC,
  title={Towards Music Captioning: Generating Music Playlist Descriptions},
  author={Keunwoo Choi and Gy{\"o}rgy Fazekas and Mark B. Sandler},
  journal={ArXiv},
  year={2016},
  volume={abs/1608.04868}
}

[2]
@article{Manco2022ContrastiveAL,
  title={Contrastive Audio-Language Learning for Music},
  author={Ilaria Manco and Emmanouil Benetos and Elio Quinton and Gy{\"o}rgy Fazekas},
  journal={ArXiv},
  year={2022},
  volume={abs/2208.12208}
}

[3]
@article{Manco2021MusCapsGC,
  title={MusCaps: Generating Captions for Music Audio},
  author={Ilaria Manco and Emmanouil Benetos and Elio Quinton and Gy{\"o}rgy Fazekas},
  journal={2021 International Joint Conference on Neural Networks (IJCNN)},
  year={2021},
  pages={1-8}
}

[4]
And arguably:
@article{Gao2022MusicQA,
  title={Music Question Answering:Cognize and Perceive Music},
  author={Wenhao Gao and Xiaobing Li and Cong Jin and Tie Yun},
  journal={2022 IEEE International Conference on Multimedia and Expo Workshops (ICMEW)},
  year={2022},
  pages={1-6}
}


Equation (4), BLEU vs. later BLUE.

**Summary Of The Paper:**

The authors propose a new task, music-to-text synaesthesia, in which audio features from musical tracks are extracted and interpreted as textual descriptions.  The authors collect a dataset composed of classical recordings and a set of manually annotated textual descriptions, and evaluate a multi-modal encoder-decoder model on the task.  They propose a topology preserving loss to propogate some learning signal from the text (the similarities between the text descriptions of different examples) to guide the learning of the music encoder, and show this loss improves the quality of text descriptions (measured in BLEU) over the baseline contrastive or triplet loss.


**Summary Of The Review:**

In favor of this paper, a new dataset of paired classical music pieces and text descriptions is collected, and a number of models are evaluated on it.  It's reasonable clear in most of its presentation, and the evaluation is probably sufficiently thorough.

On the other hand, it is questionable how novel this task really is, as defined here and on this dataset, over previous music captioning and music labeling work.  There is also minimal novelty on the modeling side, really boiling down to the topology preserving loss function.  Subjectively, studying the generated descriptions did not make a compelling case for the descriptions being more informative than having a small set of text labels (albeit necessary to have more than the single labels gleaned from the music categories).  I found the general motivation of the task confusing and overlooking previous related work.

---

> ### Author Response · Authors · 2022-11-17
> **Response to Review DjXC (Part 2/2)**
>
> **Our collected dataset**. To our best knowledge, the data from https://www.earsense.org/ might probably be the largest dataset that is suitable for our task. Ref [3] used a private production music dataset, even without the source of their private data. Given the limited information on production music released by the authors, it is unclear if these recordings support the generation of text descriptions that contain semantic meanings.
> Moreover, their music clips are of length between 30 and 360 seconds and captions contain between 3 and 22 tokens. While in our dataset, 95% of collected music pieces vary from 2.5min to 14min, with the corresponding descriptive texts ranging from 14 to 192 tokens.
>
> **Another dataset**.
> We agree that adding another dataset would be beneficial for demonstrating the generalization and effectiveness of our proposed method. Some music recordings on earsense have a second comment by another commentator. **We collect these data to be another dataset with 50 records** and validate our models on it. The BLEU score of the baseline Coordinate model is 8.54 and our GTP loss model is 8.81. Although this dataset is small, it demonstrates the capacity of our model to understand the same music recording by different people.
>
> **The descriptions don't bring something new compared to the multi-label classification while the generated text may be a few categories strung together with text.** The task we address here is to generate descriptive texts from music recordings, rather than music (multi)-labeling or tagging that predefines the category space. We believe Reviewer DJXC understands the difference between image captioning and image classification; thus, we do not need to explain more on this point. Back to this concern, the tone of Reviewer DJXC is soft and uncertain. We sincerely solicit more cues and evidence for our further exploration, which might provide more informative descriptions if this issue really exists and we can fix it.
>
> **Misc**. Thank you very much for the comments on the typo in Eq. (4).
>
> **Summary**.
> We are really appreciative of the time and efforts of Reviewer DJXC on reviewing our paper. From the comments, we sincerely feel Reviewer DJXC was professional and responsible. From the author's perspective, we are all happy to receive any criticism and more eager to follow constructive comments to tackle the criticism. However, we find some of the comments (as discussed above) are vague without further concrete suggestions for improvement, so we are not able to provide targeted responses or provide more experimental results. It is our sincere hope that Review DjXC could give more direct and constructive comments, so we could use them to help improve the quality of our paper.
>
> [1] Choi, K., Fazekas, G., McFee, B., Cho, K., & Sandler, M. (2016). Towards music captioning: Generating music playlist descriptions. arXiv preprint arXiv:1608.04868.
>
> [2] Manco, I., Benetos, E., Quinton, E., & Fazekas, G. (2022). Contrastive audio-language learning for music. arXiv preprint arXiv:2208.12208.
>
> [3] Manco, I., Benetos, E., Quinton, E., & Fazekas, G. (2021, July). MusCaps: Generating Captions for Music Audio. In 2021 International Joint Conference on Neural Networks (IJCNN).
>
> [4] Gao, W., Li, X., Jin, C., & Tie, Y. (2022, July). Music Question Answering: Cognize and Perceive Music. In 2022 IEEE International Conference on Multimedia and Expo Workshops (ICMEW).

---

> ### Author Response · Authors · 2022-11-17
> **Response to Review DjXC (Part 1/2)**
>
> Thanks for your detailed review and generous comments. Below we address our main points regarding this review.
>
> **Why not music captioning?** At the very beginning, we choose "music captioning" as the candidate, which makes the reader easier to get the point due to the well-known image captioning. However, we notice that the generative texts in image captioning are usually short. Inspired by a famous prose *Moonlight over the Lotus Pond*, where the author used synaesthesia to build the connection among vision, music, and text, we use the term "synaesthesia" to define our task. But we are also open-minded to use "music captioning" instead, if this is more exact and understandable. Please advise.
>
> **Along the lines of [2]**.
> We are a little confused about what Reviewer DjXC expects here. It would be nice that Reviewer DjXC can provide more details.
>
> **Key references**. Thank you very much for providing these key references. We are respectful of all the existing literature and would like to cite them and add a thorough discussion in our revised version. Especially for [3], a recent paper that addresses the same research problem.
> We highlight two differences.
> First, our newly developed dataset better serves the music description generation task regarding the complexity of the music and richer information in the corresponding text (discussed in detail in the following point).
> Second, we propose GTP loss, which aims at obtaining better discrimination with a group of reference samples to facilitate the transformation between different modalities, while in [3] the authors only use an encoder-decoder network consisting of a multimodal CNN-LSTM encoder with temporal attention and an LSTM decoder.
>
> Here we provide the performance of our method and MusCaps [3] in the tables below. MusCaps gets a much worse result than our methods. That's because MusCaps uses GloVe to tokenize and make word embeddings. The tokenizer cannot handle low-frequency words well which produces a lot of unknown tokens [unk]. Besides, the text decoder in MusCaps contains only a single layer of LSTM which may be difficult to fit our long and complex sentences. The official recommended sentence length of 3-22 tokens is also far below the average length of 57 tokens of our dataset. Thus we believe it's hard for MusCaps to get good performance in our dataset.
>
> |         | Split#1 | Split#2 | Split#3 | Split#4 | Split#5 | Average |
> |---------|---------|---------|---------|---------|---------|---------|
> | MUSCAPS | 0.78    | 0.79    | 1.03    | 0.58    | 0.55    | 0.75    |
> | OURS    | 6.94    | 6.71    | 6.91    | 6.37    | 6.87    | 6.76    |
>
> |||
> |---|---|
> | Case #1 | **Reference**: the rousing allegro vivace finale begins with a sweetly singing adagio introduction that very briefly seems to recall the extended introductions of the earlier op. 5 sonatas and provides the only real hint of a slow movement that will otherwise have to wait for the last cello sonata for its fully independent flowering.|
> ||**MusCaps**: the finale allegro is a [unk] of the main theme is a [unk] and [unk] [unk] with a [unk] [unk] |
> | Case #2 | **Reference**: the movement, allegro molto e con brio, is based on two subjects, the first is bright and lively while the second is dreamy with an improvisational aura. the development is ingenious and an exciting coda caps off this first rate work.|
> ||**MusCaps**: the finale allegro moderato is full of variations and [unk]|
> | Case #3 | **Reference**: the movement, allegro moderato, begins with a somewhat passionate, very characteristic quartet - like melody which leads to an ingratiating, lyrical second theme.|
> ||**MusCaps**: the finale allegro con moto is a very lyrical melody|

---

> > ### Comment · Reviewer_DjXC · 2022-12-06
> > **Synaesthesia, and Separating the task from music tagging**
> >
> > re: synesthesia
> >
> > I understand, but whether a description is long or short should not IMO have any bearing on whether something is/is not synesthesia, since synesthesia is a matter of perception, i.e., regarding how information is processed.  Although any analogy between human and computer modeling will always have some degree of misalignment, in this case, I think it is not an issue of degree -- it's just not an appropriate use of the word.  What I mean by "Along the lines of [2]" is that the concept of synesthesia seems more appropriate when two modalities are represented in the same embedding space, something which has been done previously, under the name of captioning.  This is not an issue that directly plays a part in my score, but I do personally feel that referring to the work as captioning makes it more immediately clear to the reader what the task is, which is always the goal of scientific writing.
> >
> > re: MUSCAPS
> >
> > Thank you for exploring additional related work and running additional experiments.  I think it would take a bit more care to adopt their model to your dataset if UNKs are a prevalent problem in MusCaps output, if wanting to use MusCaps as a baseline system (these embeddings are roughly 6-7 years old at this point).  I think it's obvious an output sentence with 75% unks is not really a failing of the model in a way that we should view MusCaps as somehow inherently bad at this task.
> >
> > My recommendation for a baseline that would also address concerns about the importance of this task as a task distinct from music tagging.  Use a music tagging model to predict a small set of relevant labels, adjectives/tempo/movement type, etc., for a couple segments of the audio.  Then fine-tune (or even prompt) basically any PLM to map the sequence of adjectives to a larger caption.  Something of the form (if B is to mean bars or any temporal division to capture how the performance changes over time):
> >
> > B1:sweetly/allegro/adagio; B2... B3...:  OUT: the rousing allegro vivace finale begins with a sweetly singing adagio introduction that very briefly seems to recall the extended introductions of the earlier op. 5 sonatas and provides the only real hint of a slow movement that will otherwise have to wait for the last cello sonata for its fully independent flowering.
> >
> > Then compare that model to your proposed model.  Any performance benefit of your model is then more specifically due to a better and more direct use of audio features, something missing from music tagging, and the modified loss.  A weakness of the paper in its current form is that it doesn't make clear that the information presented in the captions is information that couldn't be conveyed more succinctly with a smaller set of labels.

---

> > > ### Author Response · Authors · 2022-12-09
> > > **Re: A music tags baseline**
> > >
> > > Thank you for your suggestion. We agree with you that generating music descriptions using pre-trained language models with music tags would be an interesting baseline to explore. In our work, we have also considered a similar baseline, namely Tags (Representative) shown in the heuristics methods section. As to fill music tags into appropriate sentence templates, we select the sentence with the highest BLEU score among all sentences with the same tags as the template. This ensures that the selected sentence reads reasonable with the given tags filled in. We observe that even with ground-truth tags, this method does not generate ideal results. Using more music tags might help, however it also suffers the cost of training more classifiers.
> > >
> > > Thanks again for your time and efforts!

---

### Official Review · Reviewer_TFVv · 2022-11-02

**Confidence:** 3
**Correctness:** 2
**Technical Novelty And Significance:** 3
**Empirical Novelty And Significance:** 3
**Recommendation:** 3

**Clarity, Quality, Novelty And Reproducibility:**

The writing is mostly clear and of good quality. Below I list some issues I noticed:

- "convert Italic, French, German" -> "Italian"
- the first paragraph of "Preliminary Exploration" is written too informally.
- I am not sure why Related work is in Section 5 and not earlier in the paper.
- Missing relevant work: Won, M., Salamon, J., Bryan, N. J., Mysore, G. J., & Serra, X. (2021). Emotion Embedding Spaces for Matching Music to Stories. ISMIR 2021

The task is novel.

The code is given online, but the data isn't. I assume everything will be open after acceptance.

**Strength And Weaknesses:**

Strengths
- The paper is well written with a few exceptions. See the specific comments below.
- The motivation is clear.
- The proposed methodology is sound
- The experiments are well-prepared, however, they should be extended significantly.

Weaknesses:

- The expert descriptions exemplified in the paper encompass 1) quantifiable information (musical tempo, ...), 2) subjective information (valence, ...), and 3) writing style. I could also imagine that some contain 4) factual information, which is not possible to extract from only the music signal itself (e.g. name of the piece, composer). The proposed model seems to somewhat capture each element and is able to generate much more coherent examples than the other pre-trained models, but it also generates a lot of non-sensical descriptions such as "begins with a theme of the main theme of the main theme in the...," "the music is a fugue... to the sonata form of pizzicato," "the music of the movement."

I find similarities to models trained for question answering task, in particular, how LLMs fail to give correct answers - sometimes in a dangerously confident tone (e.g. "will my stomach be cleaner if I drink bleach?"). For me, a more interesting discussion would be about the challenges of this task, and how different models can or cannot cope with the elements mentioned above, potentially via an ablation study.

- The dataset is quite small, which probably contributes to the limitations. I appreciate that it's very difficult to collect data for such a task, however, the impact will be minor unless the data could be collected in scale. For instance, the researchers could add a second dataset consisting of Anglo-american pop music (e.g. available in Lakh dataset) and crawl expert or fan reactions (e.g. from social media) and repeat the experiments. This way we could understand how generalizable the approaches are and contrast the modes of good and bad descriptions across datasets.

- Classical pieces in the dataset are typically instrumental. It may not possible to draw out a universal (or a consensus between experts) sentiment from instrumental classical music (or any other type of music). The paper doesn't address subjectivity much in the account.

**Summary Of The Paper:**

The paper presents a novel generative task of generating textual descriptions from music. They focus on a small dataset of classical music recordings paired with expert descriptions and present comparative experiments using various pre-trained language models vs their method with a novel topology-preservation loss.

**Summary Of The Review:**

The shortcomings I listed above limit the work's scope. Therefore, I believe the paper is more suitable for a music information retrieval conference like ISMIR at its current stage.

---

> ### Author Response · Authors · 2022-11-17
> **Response to Review TFVv**
>
> Thanks for your detailed review and generous comments. Below we address our main points regarding this review.
>
> **Non-sensical descriptions**. We agree with Reviewer TFVv that music and text are two different modalities, which cannot contain exact the same information. In general, they might share a lot in common, which is the assumption of our task. Currently, our generative texts suffer from non-sensical descriptions. We conjecture it might result from the limited dataset, rather than the language generative model. We believe LLM means large language model. Feel free to correct us if our understanding is incorrect. In our preliminary attempt at the beginning of this project, we have used large pre-trained language models like BART [1] and not got ideal results. From the illustrative example in Figure 1, the music corresponding texts have a very different distribution compared to the general domain corpus. Beyond the ablation on different backbone models, feel free to kindly educate us on what extra experiments we need to further verify.
>
> **Why not crawl fan reactions from social media?** We are really thankful for this valuable comment. We feel Reviewer TFVv was standing in our shoes and provided potential solutions for further improvements, rather than simply criticizing without any supporting evidence.
>
> Actually, crawling comments from music sites or social media is our initial idea. We found that most comments are judgments like "sounds good" or fan words like "I like the singer," which are not descriptions of the music itself. Some more complicated comments are usually listeners' stories related to the music. However, we need descriptions of music content or sentiment for our task. To our best knowledge, the data from https://www.earsense.org/ might probably be the largest dataset that is suitable for our task. Beyond reaction, if Reviewer TFVv knows better datasets, we are eager to try our model on a large dataset, which might solve the Non-sensical issues to some extent.
>
> **Another dataset**.
> We agree that adding another dataset would be beneficial for demonstrating the generalization and effectiveness of our proposed method. Some music recordings on earsense have a second comment by another commentator. **We collect these data to be another dataset with 50 records** and validate our models on it. The BLEU score of the baseline Coordinate model is 8.54 and our GTP loss model is 8.81. Although this dataset is small, it demonstrates the capacity of our model to understand the same music recording by different people.
>
> **The classical pieces are typically instrumental with subject sentiment.**
> Subjectivity is really an important problem for music understanding. There are two kinds of subjectivity. One is that the same sentiment may be described by different words which were discussed in Section 4.2. Another is that the same music may draw out different sentiments which is the issue of this review. Classical music eliminates this problem because most classical music follows composition principles which make it convenient to analyze and possible to meet consensus in sentiment.
>
> **Misc**. Thank you very much for the comments on the typo, organization, writing tone, and key reference. We have modified them accordingly and would like Reviewer TFVv to read them again. Hopefully, this time will have a better experience.
>
>
> [1] Lewis M, Liu Y, Goyal N, et al. "BART: Denoising Sequence-to-Sequence Pre-training for Natural Language Generation, Translation, and Comprehension", Proceedings of the 58th Annual Meeting of the Association for Computational Linguistics. 2020.

---

### Official Review · Reviewer_ahJ7 · 2022-11-04

**Confidence:** 3
**Correctness:** 3
**Technical Novelty And Significance:** 3
**Empirical Novelty And Significance:** 2
**Recommendation:** 5

**Clarity, Quality, Novelty And Reproducibility:**

The paper is relatively well written but I have some comments/questions which follow in the end of this section, I think addressing those would make the paper much clearer.

The paper is novel but quite application oriented, and explores only a small dataset.

The authors have provided code.

__Some comments / questions for clarity__

-  I am not sure whether introducing the term "synaesthesia" helps or whether it confuses the reader, since such tasks are already known in the literature (e.g. captioning for images). Nevertheless, as far as I'm aware this task has not been tackled for music, although I am not very familiar with the applications of generative models in the music domain.
- Three sentences before sec. 3: Can you elaborate, I don't fully follow.
- It would help to mention earlier on in the text that the raw music spectrum used also captures some temporal aspects as it is made clearer in "Implementation"
- Second bullet in "Comparison models": Can you be a bit clearer what are the pre-trained parts of those methods and where / how they have been pre-trained?
- Second sentence in 4.2: Can you clarify further? I am not sure I understand what "low performance" refers to and what is the referred "increased difficulty for humans...".
- Table 1: What are the limits (smallest/largest possible) for the scores presented here?


**Strength And Weaknesses:**


__Strengths:__
* Good motivation
* New dataset
* Reasonable learning component
* GTP Loss
* Results generally positive
* Cross validation and small variance

__Weaknesses:__
* Small dataset - how to train deep learning models?
* Little methodological contribution
* Not entirely convinced of the GTP Loss
* Results not very exciting and only on a single experiment/dataset
* Some clarity needed with presentation of scores
* Generated sentences not always syntactically sensible, to some extent defeats the purpose of moving beyond tag generation.
* Sentiment analysis results lack baselines

__Details:__

The motivation to extend music-based text generation beyond tags is interesting, with an underpinning effort to communicate the various nuances found in music using language, a widely understood modality for humans.

The dataset collection is a reasonable and useful step in this work. The collected dataset unfortunately is quite small and specific to a specific music type. I was actually quite surprised that the authors managed to train multi-layer, multi-modal models with less than 2K recordings. I understand a pre-trained language model was used (i.e. on English language - hope the authors can correct me if I misunderstood), would it be possible to also use a pre-trained music model instead of pre-training it on the actual collected (small) dataset? For example, by leveraging other public, even if unlabelled, music recording databases.

The learning component of the approach is quite straightforward and reasonable. The autoencoders learn representations for each modality, topped by a mapping encoder. The authors also introduce the so-called group topology-preservation (GTP) loss. I find this generally reasonable as intuition, but I'm not entirely convinced that it has been adequately justified and explored, especially being the main methodological contribution.

Firstly, I wonder whether such constrain forces information in music which is not present in text to unnecesarily be aligned in a particular way, thus making the model less flexible. In other words, the GTP constrain is an indirect way of obtaining better discrimination but has additional effects which should be explored to understand them better, i.e. whether they introduce problems with generalization. Perhaps running experiments in different datasets with this loss would shed some light.

Secondly, I wonder how the GTP loss compares to (cross-modality) triplet loss if we were to increase the number of negative (and perhaps also positive) samples of the triplet loss. Is it correct to say that for adequately many samples the two losses become very similar?

Given the absence of studies on the same task the authors propose a number of baselines which I find reasonable to use as competitive methods. The proposed method seems to perform well compared to tag-based approaches which is very encouraging. I had some trouble interpreting the results of the table though because I am not sure what are the minimum and maximum values of the score, e.g. the authors mention BLEU but I see scores > 1. It seems to me that the GTP loss is not introducing a very significant boost the performance.

The qualitative results seem interesting but the generated sentences are not always syntactically sensible. This to some extent defeats the purpose of moving beyond tags, but on the other hand with a larger dataset it might have been less of an issue. I also wonder if adapting large language models would solve the issue.

I appreciate that the authors perform cross-validation given the small set and I think the small variance in the results is encouraging. I also enjoyed the transfer ability of sentiment, although I wonder why the results of this are not shown for the baselines.


**Summary Of The Paper:**

The authors consider the problem of generating text describing a piece of chamber music, given the acoustic features of the piece. The proposed model is relying on a text autoencoder, an audio autoencoder and a cross-modality translation model. The authors also develop the so-called group topology-preservation loss for ensuring that the representations in the music domain lie in a topology constrained by that of the text domain, which is more discriminative. The model is trained and evaluted on a dataset collected by the authors.



**Summary Of The Review:**

Overall the paper is well-motivated, the main application idea is original and the execution is reasonable. However the main positive points of the paper have to do with the application itself, with little methodological contribution/analysis especially regarding the GTP loss. There is also little room for experimentation since all the experiments are done in the small collected dataset which was also used for training. Given the above application-oriented way of conducting this research, although I feel there could be audiences within ICLR finding this useful, the paper would be better suited to a more domain-specific venue.

Edit after rebuttal: I thank the authors for providing detailed answers to my questions. After the rebuttal period and also seeing reviewer DjXC's concerns I would still like to keep my score.

---

> ### Author Response · Authors · 2022-11-17
> **Response to Reviewer ahJ7 (Part 1/2)**
>
> Thanks for your detailed review and generous comments. Below we would like to address your main points regarding this review.
>
> **Deep learning models on small datasets.**
> We also notice the possible overfitting problem of deep learning on small datasets. To tackle this, we use pre-trained models for parameter initialization. Specifically, we first employ the pre-trained encoder-decoder models for music and text, respectively (see the blue and green lines in Figure 3), then fine-tune the music encoder and text decoder with our GTP loss to achieve generating descriptive texts from music recordings. In the paragraph on implementation, we demonstrate the satisfactory performance of pre-trained music and text models. And we did not notice any overfitting issues by comparing the performance on the training set (4 folds) and test set (1 fold).
>
> **Pre-trained music model on other large music corpora.**
> Yes. Using a pre-trained music model on other large music corpus makes our task easier, which actually was one of our attempts at the beginning of this project.
> We have considered using pre-trained music tagging models [1] to extract music features. Unfortunately, the generated descriptions were much worse with all the baseline models. We conjecture that these CNN-based models are not suitable for our collected classical music.  Thus, we trained an auto-encoder to check the representation quality through reconstruction. The average music length in our dataset is 7.4 mins, which means the auto-encoder was trained in 242 hours of music pieces.
> With visualized reconstruction and better generation results, we believe our pre-trained music model can be used for the generation task.
>
> **Not entirely convinced of the GTP loss.**
> We are more than cheerful to receive the insightful comments with detailed explanations and suggestions. This makes our response directed and targeted.
>
> (a) *Motivation of GTP*. We agree with Reviewer ahJ7 that the whole multi-modality translation framework we borrow from the coordinate model is intuitive. During the in-depth exploration, we noticed the high similarities of the music representation from the pre-trained model due to the nature of classical music. To tackle this, we have tried pairwise, triplet, and contrastive loss to increase the discrimination. Unfortunately, they did not work well (See Figure 6). This motivated us to propose the GTP loss for obtaining better discrimination with a group of reference samples.
> Personally, we are reluctant to propose any new techniques if some existing ones can well handle the research problem. Moreover, we value the contributions in all aspects, but put more weights on the research problem and philosophy than the technical part.
>
> (b) *Another datasets*.
> Some music recordings on earsense have a second comment by another commentator. **We collect these data to be another dataset with 50 records** and validate our models on it. The BLEU score of the baseline Coordinate model is 8.54 and our GTP loss model is 8.81. Although this dataset is small, it demonstrates the capacity of our model to understand the same music recording by different people.
>
> (c) *Increase the number of pairs in triplet loss*.
> We try different pair numbers $k$ in triplet loss on Coordinate model. Note that for $k \geq 16$ we employ a momentum encoder to reduce the computation budget as the implementation of GTP loss in our paper.
> Other hyperparameters e.g. margin=0.1 and weight=1 follow the best setting of $k=1$. The results seem to be constant as $k$ increases. However, there are still many hyperparameters and settings that are not well explored.
>
> |                | Split #1  | Split #2  | Split #3  | Split #4  | Split #5  | Average |
> |----------------|-----------|-----------|-----------|-----------|-----------|---------|
> | COORDINATE     | 6.41±0.34 | 6.70±0.17 | 6.20±0.21 | 6.10±0.11 | 6.66±0.24 | 6.42    |
> | TRIPLET(K=1)   | 6.60±0.14 | 6.47±0.13 | 6.74±0.07 | 6.07±0.12 | 6.95±0.13 | 6.56    |
> | TRIPLET(K=2)   | 6.02±0.14 | 6.40±0.04 | 6.40±0.15 | 5.84±0.21 | 6.55±0.25 | 6.24    |
> | TRIPLET(K=4)   | 6.20±0.16 | 6.42±0.22 | 6.78±0.14 | 6.01±0.13 | 6.62±0.27 | 6.41    |
> | TRIPLET(K=16)  | 6.16±0.07 | 6.65±0.25 | 6.32±0.15 | 5.90±0.12 | 6.61±0.29 | 6.34    |
> | TRIPLET(K=32)  | 6.36±0.28 | 6.59±0.07 | 6.50±0.20 | 5.94±0.24 | 6.71±0.21 | 6.42    |
> | TRIPLET(K=64)  | 6.23±0.12 | 6.48±0.22 | 6.27±0.08 | 6.16±0.13 | 6.35±0.29 | 6.30    |
> | TRIPLET(K=128) | 6.29±0.23 | 6.83±0.17 | 6.22±0.34 | 6.08±0.16 | 6.58±0.17 | 6.40    |
>
> **Why BLEU scores > 1 ?**
> The BLEU score reflects the similarity between the candidate text and the reference text.
> In this work, we follow the tradition of the machine translation community for reporting the BLEU score in a range of 0-100 [2, 3, 4].
> The typical BLEU score for machine translation tasks is from 10 to 40. We will clarify this in our paper.

---

> ### Author Response · Authors · 2022-11-17
> **Response to Reviewer ahJ7 (Part 2/2)**
>
> **GTP loss is not introducing a very significant boost**.
> The goal of GTP is to increase better discrimination of music representation, which we expect to bring in performance improvement accordingly. Actually, GTP indeed introduces a significant boost. With 5-fold experiments with 5 different seeds, the statistical significance p-values of comparing other methods and ours are below 0.001, which strongly supports the boost of GTP loss.
>
> **Large language models**. This was our another attempt at the beginning of this project.
> We have used large pre-trained language models like BART [4] and not gotten ideal results. We conjecture that the music-corresponding texts have a very different distribution compared to the general domain corpus.
>
> **Results of sentiment for baselines**. We put the sentimental experiments in the in-depth analysis, which only contains the exploration of our method. Here we are happy to demonstrate the results of sentiment for other baseline models. The table below shows our model has better performance in some minority categories.
>
> | | joy| neutral | sadness | fear | anger | surprise | disgust |
> |---|---|---|---|---|---|---|---|
> | Reference| 1054 | 318|277|182|60|55|9|
> | Ours | 1132 | 550| 132| 121| 12 | 3| 5 |
> | Coordinate | 1238 | 504| 117| 86| 3| 3| 4 |
> | kNN| 763| 719| 313| 89| 20 | 51 | 0 |
>
> **The term "synaesthesia."**
> At the very beginning, we choose "music captioning" as the candidate, which makes readers easier to get the point due to the well-known image captioning. However, we notice that the generative texts in image captioning are usually short. Inspired by a famous prose *Moonlight over the Lotus Pond*, where the author used synaesthesia to build the connection among vision, music, and text, we use the term "synaesthesia" to define our task. But we are also open-minded to use "music captioning" instead if this is more exact and understandable. Please advise.
>
> **Three sentences before Section 3 are confusing**.
> Sorry for the confusion. We would like to express the music representations are of high similarity, while the text representations are more discriminative than the music ones. Therefore, we aim to employ the text to guide the music representation generation by our proposed GTP loss. We have modified them as follows:
> "Thus, a natural idea is to employ the text to guide the learned music representations to make them more discriminative, so the cross-modality transformation would be easier.
> The research question we explore in this paper is: How can we employ the music-text pairs to learn the informative representation for music pieces and achieve music-to-text synaesthesia?"
>
>
> **Organization of Implementation**.
> Sorry for the confusion. We would like to follow your suggestions to mention temporal aspects of the spectrum earlier as follows: "From the music side, the original classical music is first converted into a spectrogram which has a frequency axis of 256 dimensions and a flexible temporal axis.''
>
>
> **More details about pre-training on comparison models**.
> Sorry for the confusion. We first pre-train music and text auto-encoders to get a basic encoder and decoder for each modality. Figure A3 in the appendix shows the structure schema where all the 3 pre-trained methods are mainly made up of the 4 basic pre-trained encoders or decoders. Only the discriminator in the Joint model is not pre-trained.
>
> **Second sentence in Section 4.2*.
> Sorry for the confusion. We will revise it to be the following: "Note that the ground text descriptions for classical music are sometimes hard to digest even for humans (see the cases in Table 2), it is thus not totally surprising that the performance of our model is generally low."
>
> **Smallest/largest possible in Table 1**.  The range of BLUE score is from 0 to 100. Please also see our response to "Why BLEU scores $\geq$ 1."
>
>
> [1] Minz Won, Andres Ferraro, Dmitry Bogdanov, and Xavier Serra, "Evaluation of CNN-based Automatic Music Tagging Models'', arXiv preprint arXiv:2006.00751, 2020.
>
> [2] Ashish Vaswani, Noam Shazeer, Niki Parmar, Jakob Uszkoreit, Llion Jones, Aidan N. Gomez, Łukasz Kaiser, and Illia Polosukhin. 2017. "Attention is all you need'', Proceedings of the 31st International Conference on Neural Information Processing Systems. 2017.
>
> [3] Mike Lewis, Yinhan Liu, Naman Goyal, Marjan Ghazvininejad, Abdelrahman Mohamed, Omer Levy, Ves Stoyanov, and Luke Zettlemoyer. "BART: Denoising Sequence-to-Sequence Pre-training for Natural Language Generation, Translation, and Comprehension'', Proceedings of the 58th Annual Meeting of the Association for Computational Linguistics. 2020.
>
> [4] Urvashi Khandelwal and Angela Fan and Dan Jurafsky and Luke Zettlemoyer and Mike Lewis. "Nearest Neighbor Machine Translation'', Proceedings of the International Conference on Learning Representations. 2021.

---

### Decision · Program_Chairs · 2023-01-20

**Decision:**

Reject

**Justification For Why Not Higher Score:**

Based on the reviewers' comments, I think it is very much unanimously agreed that this paper still needs a bit more work before it's ready for publication. One reviewer also suggests that a more domain-specific venue might be a good option.

**Justification For Why Not Lower Score:**

N/A

**Metareview: Summary, Strengths And Weaknesses:**

Summary: This paper considers the following research problem of generating descriptive texts from music recordings. Given that this task is beyond the capability of the current existing music-related datasets, the authors collect a new dataset which contains paired classical music recordings and text descriptions. The proposed model is a cross-modality translation model, which consists of a text autoencoder and an audio autoencoder. A group topology-preservation (GTP) loss is developed for employing the text (which is more discriminative) to guide the music representation generation. Experimental results are presented on the collected dataset.

Strengths: All the reviewers agree that the motivation of the paper is very interesting and it is great to have a new dataset for this particular problem. Overall the proposed method makes intuitive sense (albeit still with some confusions on execution as pointed out by reviewer ahJ7).

Weaknesses: There are a few major weaknesses which make accepting this paper harder to justify: 1) Currently the paper has relatively little methodological contribution. As reviewer ahJ7 pointed out, the proposed GTP loss makes intuitive sense, but is not adequately justified and explored; 2) As multiple reviewers pointed out, the dataset collected by the authors is quite small and it consists entirely of classical music which limits its applicability; 3) Multiple reviewers found the use of the term "synaesthesia" misleading given that this paper can also be descrbied as "music captioning" where some existing literature can also be included/considered. 4) The experimental setup seems reasonable but the baselines all seem pretty weak to offer much meaningful information.